# Towards Space Deployment of the NDSA Concept for Tropospheric Water Vapour Measurements

Luca Facheris [1,2] , Andrea Antonini [3,4] , Fabrizio Argenti [1] , Flavio Barbara [5] , Ugo Cortesi [2,5,*] , Fabrizio Cuccoli [2] , Samuele Del Bianco [2,5] , Federico Dogo [6,7] , Arjan Feta [5] , Marco Gai [2,5] , Anna Gregorio [6,7] , Giovanni Macelloni [2,5] , Agnese Mazzinghi [1,2] , Samantha Melani [3,4] , Francesco Montomoli [2,5] , Alberto Ortolani [3,4] , Luca Rovai [3,4] , Luca Severin [6,7] and Tiziana Scopa [8]

1. Department of Information Engineering, University of Florence, 50139 Florence, Italy; luca.facheris@unifi.it (L.F.); fabrizio.argenti@unifi.it (F.A.); agnese.mazzinghi@unifi.it (A.M.)
2. National Inter-University Consortium for Telecommunications (CNIT), 43124 Parma, Italy; fabrizio.cuccoli@unifi.it (F.C.); s.delbianco@ifac.cnr.it (S.D.B.); m.gai@ifac.cnr.it (M.G.) g.macelloni@ifac.cnr.it (G.M.); f.montomoli@ifac.cnr.it (F.M.)
3. Istituto per la BioEconomia del Consiglio Nazionale delle Ricerche (IBE-CNR), 50019 Sesto Fiorentino, Italy; antonini@lamma.toscana.it (A.A.); melani@lamma.toscana.it (S.M.); ortolani@lamma.toscana.it (A.O.); rovai@lamma.toscana.it (L.R.)
4. LaMMA Consortium, via Madonna del Piano 10, 50019 Sesto Fiorentino, Italy
5. Istituto di Fisica Applicata Nello Carrara del Consiglio Nazionale delle Ricerche (IFAC-CNR), 50019 Sesto Fiorentino, Italy; f.barbara@ifac.cnr.it (F.B.); arjanfetahu@gmail.com (A.F.)
6. Department of Physics, University of Trieste, 34127 Trieste, Italy; federicodogo@picosats.eu (F.D.); anna.gregorio@units.it (A.G.); lucaseverin96@gmail.com (L.S.)
7. PICOSATS S.R.L., 34149 Trieste, Italy
8. ASI—Agenzia Spaziale Italiana, 00133 Rome, Italy; tiziana.scopa@asi.it
* Correspondence: u.cortesi@ifac.cnr.it; Tel.: +39-055-5226368

**Abstract:** A novel measurement concept specifically tuned to monitoring tropospheric water vapour's vertical distribution has been demonstrated on a theoretical basis and is currently under development for space deployment. The NDSA (Normalised Differential Spectral Attenuation) technique derives the integrated water vapour (IWV) along the radio link between a transmitter and a receiver carried by two LEO satellites, using the linear correlation between the IWV and a parameter called spectral sensitivity. This is the normalised incremental ratio of the spectral attenuation at two frequencies in the Ku and K bands, with the slope of the water vapour absorption line at 22.235 GHz. Vertical profiles of WV can be retrieved by inverting a set of IWV measurements acquired in limb geometry at different tangent altitudes. This paper provides a comprehensive insight into the NDSA approach for sounding lower tropospheric WV, from the theoretical investigations in previous ESA studies, to the first experimental developments and testing, and to the latest advancements achieved with the SATCROSS project of the Italian Space Agency. The focus is on the new results from SATCROSS activities; primarily, on the upgrading of the instrument prototype, with improved performance in terms of its power stability and the time resolution of the measurements. Special emphasis is also placed on discussing tomographic inversion methods capable of retrieving tropospheric WV content from IWV measurements, i.e., the least squares and the external reconstruction approaches, showing results with different spatial features when applied to a given atmospheric scenario. The ultimate goal of deploying the NDSA measurement technique from space is thoroughly examined and conclusions are drawn after presenting the results of an Observing System Simulation Experiment conducted to assess the impact of NDSA data assimilation on environmental model simulations.

**Keywords:** tropospheric water vapour; total column and vertical profile; active microwave sounding; Normalised Differential Spectral Attenuation; assimilation methods

## 1. Introduction

Tropospheric water vapour (WV) constitutes about 99% of the total WV in the Earth's atmosphere. The amount of water available for precipitation in any form depends on the global and local dynamics of WV; in addition to that, WV plays a role in atmospheric transport processes since it influences the atmospheric thermal profiles through latent heat exchange processes related to its phase changes (from gas to liquid and vice versa). Consequently, measuring the concentration, the spatial distribution (both vertical and horizontal), and the variation with time of WV is of paramount importance in meteorology and climatology [1–3]. Despite the efforts, the challenge of how to achieve such an objective with the appropriate resolution and sensitivity is still open [4–6].

The most precise instruments and methods to measure WV are also the most expensive ones: we refer to radiosondes or probes, measuring in situ the local content of humidity, and to lidars [7]. For evident reasons, these solutions cannot be proposed for the global and continuous monitoring of WV, which can be pursued only by resorting to satellites. The systems on board exploit electromagnetic signals, which, being sensitive—at one or multiple frequencies—to the presence of WV, make it possible to estimate its amount by converting some parameters of such signals. Here, we have a variety of systems and methods: for instance, methods based on passive systems (radiometers) working in the infrared spectrum [8] (even from geostationary satellites—see [9]) or on active ones, exploiting signals from global navigation satellite systems (GNSS). The latter is the case for the Radio Occultation (RO) methods, which provide, with the aid of independent auxiliary information, average WV vertical profiles over an area through inversion methods applied to such signals [10]. As a matter of fact, GNSS signals received at ground stations give additional and unprecedented opportunities to get WV estimates above a network of receivers [1], possibly using particular tomographic approaches [11–13]. However, such estimates cannot be provided over seas and oceans. Among the active systems, we also mention spaceborne radars: through differential absorption measurements, they can provide the columnar WV content, which is a key parameter in numerical weather prediction (NWP) models [14].

A common problem of all the global monitoring systems available at present is the accuracy issue of the WV measurements in the lower troposphere where, on the other hand, the WV concentration is at the maximum [15].

Years ago, some of the authors of this paper proposed to exploit the normalised incremental ratio of spectral attenuation at frequencies in the Ku and K bands (specifically, from 17 to 21 GHz) to estimate the tropospheric integrated water vapour (IWV), intended as the absolute mass of water vapour along the path between a couple of Low Earth Orbit (LEO) satellites. It is assumed that the first one carries a transmitter and the other one a receiver and that measurements are made in a limb measurement geometry [16]. The reason for using those frequencies is their proximity to the WV absorption line. There, the slope of the absorption spectrum depends only on the WV content along the radio link, as the contributions of the other atmospheric components are practically constant with the frequency.

The aforementioned normalised incremental ratio has been referred to as 'spectral sensitivity'. It was demonstrated that the spectral sensitivity at a given frequency is linearly correlated to the IWV along the radio link placed at a given tangent altitude (TA), the minimum tropospheric altitude of the link. Specifically, at the lower and higher end of the 0–10 km TA interval, the highest correlation between spectral sensitivity and IWV is found at the lower and higher end of the 17–21 GHz frequency range, respectively. Therefore, in the absence of signal impairments, accurate estimates of the IWV along a LEO-LEO satellite can be achieved by converting the measured spectral sensitivity into IWV through a multi-frequency approach, i.e., by using different frequencies for different TA intervals. The conversion to IWV is then easily made through predetermined linear relations, depending on the TA and frequency [16,17].

This method to estimate the IWV along a radio link between two LEO satellites has been referred to as Normalised Differential Spectral Attenuation (NDSA). It was originally proposed with the aim of probing the tropospheric WV by means of a pair of counter-rotating LEO satellites (i.e., moving in opposite directions in the same orbital plane), performing IWV measurements during a relative rise or set event of the satellites and thereby providing vertical IWV profiles [16]. From the latter, the vertical profiles of WV (intended as absolute humidity) can in turn be estimated through inversion techniques. The potential of the NDSA technique for tropospheric water vapour sounding from counter-rotating LEO satellites was demonstrated by a number of theoretical studies supported by the European Space Agency (ESA) (ALMETLEO, 2003–2004 [18]; ACTLIMB, 2008–2010 [19]; ANISAP, 2012–2013 [20]). Such studies focused also on the performance analysis in the presence of impairments [21]. Particular attention was given to the analysis of scintillation induced by tropospheric turbulence, showing that the differential approach used by the NDSA technique limits its negative effects [22]. As to the presence of liquid water (LW) along the radio path, which biases the IWV estimates, it was shown that an additional spectral sensitivity measurement at 32 GHz is useful to detect LW, and possibly to correct the bias error induced by LW on the IWV estimates [17].

In the ESA-ANISAP study, it was envisaged that the NDSA principle could be profitably used not only in a counter-rotating configuration of LEO satellites, but also in a co-rotating one, where a constellation of satellites still orbit in the same plane, but move along the same direction. Figure 1, for instance, shows the case of four satellites with a transmitter on board, which are always in view of four satellites with a receiver on board: therefore, we have sixteen tropospheric links (only six are shown for the sake of simplicity) and, using the NDSA technique, we can get sixteen simultaneous measures of IWV at every integration interval. As a result, in general, there is a number of constantly active radio links, which cross the troposphere at fixed TAs, scanning an annular region of the orbital plane and getting a sequence of IWV measurements. The minimum and maximum altitude of such annular regions are defined by the two microwave links with the minimum and maximum TA, respectively (for the sake of simplicity, we assume a spherical Earth so that the tangent altitude of each link does not vary). Based on the latter measurements, tomographic inversion schemes can thus provide WV concentration fields over planes perpendicular to the Earth's surface (see the yellow box sketched in Figure 1) to retrieve the WV field.

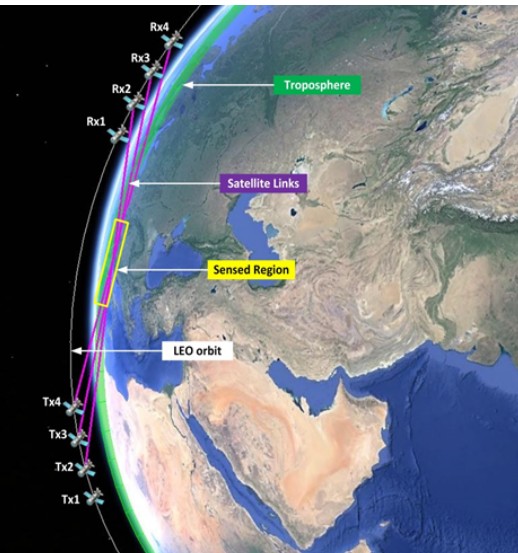

**Figure 1.** Acquisition scheme of the NDSA measurements of IWV for a constellation system of co-rotating LEO satellites.

This novel measurement concept was analysed in better detail in a 2016 paper [23] and the first practical implementation of the NDSA measurements was possible thanks to the SWAMM (Sounding Water vapour by Attenuation Microwave Measurements—2016–2018) project, financed with funds from the Tuscany region of Italy. During SWAMM, the first prototype was built with a low-cost instrument capable of providing spectral sensitivity measurements at 19 GHz. A measurement campaign was carried out along a ground-to-ground link of about 60 km, whose results have been recently published [24].

These results and activities revealed the path to a more in-depth investigation of the overall actual potential of the NDSA technique applied to the co-rotating satellites case, which involved not only the processing methods to retrieve the WV concentration fields on a global scale and continuous time basis, but also the impact of the NDSA products on meteorological forecasts, and a payload/mission analysis. Such a pre-feasibility study of a space mission for WV monitoring, based on a constellation of satellites and on the cost-affordable CubeSat technology, has been funded by the Italian Space Agency (ASI) and called SATCROSS. In this work, we summarize the SATCROSS outcomes and delineate the open issues and perspectives for further developments towards the deployment of the NDSA concept for on-board co-rotating satellite platforms.

The paper is structured as follows: in Section 2.1, the NDSA concept is briefly described and the main results of the related studies are recalled. Section 2.2 describes the prototype instrument that has been developed for performing NDSA measurements and the campaigns in which this instrument has been involved so far. Section 3 introduces the tomographic inversion methods that have been developed for retrieving the two-dimensional water vapour fields from sets of IWV measurements made by ensembles of co-rotating LEO satellites, while Section 4.3 is devoted to the analysis of the potential of such measurements when assimilated in NWP models. Section 4.1 shows how WV estimates can be biased by the presence of liquid water along the microwave links and how such bias can be effectively corrected or—at least—the presence of liquid water detected. The possible solutions for a space mission like that envisaged by the SATCROSS project team, based on CubeSat technology, are described in Section 4.2. Conclusions and future perspectives are discussed in Section 5.

## 2. Briefings of the NDSA Concept and Instrument Prototype

In this section, we briefly recall the fundamentals of the NDSA approach to total water vapour mass estimation, which is present along a radio link. Then, we describe the first instrumental implementation of such an approach. We do this by recalling the results achieved step by step through the ESA, SWAMM, and SATCROSS studies involving the estimation of a parameter depending on the differential attenuation at two frequencies.

### 2.1. NDSA Concept

The NDSA technique is based on the conversion of a spectral parameter, the *spectral sensitivity S*, measured along the propagation path between a transmitter and a receiver, into the corresponding path-integrated WV through simple pre-set linear IWV-*S* relations. As detailed in [16], where the transmitter and receiver were supposed to be located on board two separate LEO satellites, the spectral sensitivity at a given frequency is the finite difference approximation of the spectral attenuation function (i.e., the total attenuation undergone by a tone signal during its propagation) at that frequency. Therefore, one method of measuring $S$ at $f_o$ is to transmit two sinusoidal tones at relatively close frequencies $f_1$ and $f_2$ ($f_1 > f_2$), placed symmetrically around $f_o$, which is referred to as the channel frequency. At the receiver, through appropriate filtering, the powers $P_1$ and $P_2$ corresponding to each of the two received tones are measured, so that the spectral sensitivity attributable to $f_o$ is simply derived as

$$S_{f_o} = \frac{P_2 - P_1}{\Delta f P_2} \tag{1}$$

where $\Delta f = f_1 - f_2$ is the spectral separation, whose most appropriate values are 200 or 400 MHz.

The proposal to use such a normalised differential approach, which is intrinsic to the definition of spectral sensitivity, dates back to the ESA ACE+ mission studies [25], funded under the Second Call for ESA's Earth Explorer Opportunity Missions. Such studies highlighted the heavy impact that scintillation due to tropospheric turbulence may have on WV estimates provided by limb measurements between two LEO satellites performed in the aforementioned frequency range. Thanks to its normalised incremental approach, the NDSA method appeared as a convenient alternative to those envisaged within ACE+ to limit this impact. The idea, later confirmed (see [22]), that the tropospheric scintillation had a reduced impact on the spectral sensitivity is mainly based on the hypothesis that the correlation between the fluctuations due to scintillation on the two reception channels is very high, since the frequencies of the two tones are relatively close.

The very high correlation between $S_{f_o}$ and IWV at different TAs for $f_o \in [17–21]$ GHz has been verified by simulations based on a large set of radio sounding data [16]. This was a very important further stimulus to study the other aspects of the applicability of the NDSA method to the connection between two counter-rotating LEO satellites, which led to an independent ESA study (AlMetLEO: Alternative Measurements Techniques for LEO-LEO Radio Occultation [18]). The AlMetLEO outputs were: (1) the characterization, in a parametric way, of the IWV-$S_{f_o}$ relationships at various TAs (up to 12 km, for the corresponding 'optimal' value of $f_o$), based on an extensive analysis of the radio sounding data; (2) the model of the signal fluctuations due to tropospheric turbulence; (3) the estimation of the accuracy of the spectral sensitivity measurements achievable with given SNR and scintillation power [21].

The subsequent ESA-ACTLIMB study (Study of the Performance Envelope of Active Limb Sounding of Planetary Atmospheres, [19]) allowed for the deepening of other aspects of the NDSA technique, opening further perspectives for its use in a multifrequency context. In particular, the potential of the 179 to 182 GHz frequency band to estimate IWV at tangent altitudes greater than 10 km clearly emerged. This frequency band also features a significant robustness to fluctuations due to scintillation.

As a matter of fact, the IWV along a given microwave link at $f_o$ can be obtained from linear relations:

$$IWV = a_o S_{f_o} + b_o \tag{2}$$

In [16,17] are reported the 'optimal' values of $f_o$ as functions of different intervals of tangent altitudes, and the corresponding values of $a_o$ and $b_o$.

Finally, the ESA-ANISAP study (Analysis of the NDSA technique for Inter-Satellite Atmospheric profiling)[20] provided a complete and definitive view of all the theoretical aspects related to the NDSA technique applied to the case of two counter-rotating LEO satellites in a context in which the goal is to recover the vertical profiles of IWV and/or WV. This included the possibility of using the estimates of $S_{f_o}$ with $f_o = 32$ GHz in order to detect the presence of LW along the link and correct through these estimates for the bias caused to the WV estimates made in the 17–21 GHz frequency range.

Among the objectives of the ESA-ANISAP study, there was an ancillary one: to understand the potential of the NDSA technique applied to the case of a set of co-rotating LEO satellites, i.e., the case in which one or more satellites with a transmitter on board follow, in the same orbit and along the same direction, one or more receiving satellites, as mentioned in the introduction and shown in Figure 1. Though this novel concept could be only superficially analysed during the study, it seemed quite attractive, since it implied the possibility to estimate the two-dimensional (2D) distribution of WV concentration instead of single vertical profiles, as in the counter-rotating case. Furthermore, a constellation of co-rotating satellites, whose set up is nowadays possible with affordable costs thanks to the CubeSat technology, allows integration intervals longer than the counter-rotating configuration (where the integration time is limited by the speed of the relative rises

and sets of the two LEO satellites), with benefits in terms of the reduction of noise and scintillation at the receiver.

These basic ideas were consolidated later on in a work focused on the definition of an inversion technique for retrieving 2D WV fields from sets of IWV measurements collected during the observation time through the NDSA technique [23]. The work, jointly with the past NDSA experience, has been the key to get the support of the Italian Space Agency for the SATCROSS project, which has been fully devoted to the analysis of the different aspects involved in the co-rotating satellites approach. The SATCROSS activities described in the next sections focused on the design and development of the key elements of a spaceborne mission onboard a constellation of small satellites: from building and testing an upgraded prototype of the NDSA instrument, to a more in-depth analysis of appropriate algorithms for tomographic inversion, to the investigation of potential impact of NDSA WV products on NWP models, to the definition of the mission characteristics and of the satellites payload, based on CubeSat technology.

### 2.2. Instrument Prototype

In the context of the SWAMM project [24], to demonstrate the feasibility of the NDSA technique, we designed and implemented a first demonstrator (SWAMM-1.0) of such technique operating in a ground-to-ground link configuration [24]. The selected central frequency $f_o$ was 19 GHz and the $\Delta f = 400$ MHz such that $f_1 = 18.8$ GHz and $f_2 = 19.2$ GHz. Along with the data obtained from its measurement campaign, SWAMM-1.0 led in the right direction for demonstrating the NDSA theoretical foundations. Nevertheless, its characteristics of measurement accuracy and sensitivity were not sufficient for a demonstrator of a proper level, and certainly incompatible with possible developments in future ground-air applications. The first prototype evolved in an upgraded version, hereafter referred to as SWAMM-2.0, with the realization of improved transmitter and receiver units to ensure more accurate IWV measurements; its architecture is illustrated in Figure 2.

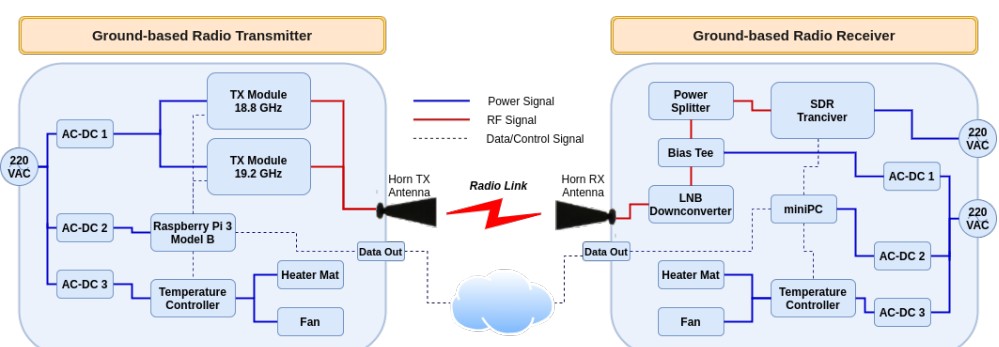

**Figure 2.** SWAMM-2.0 system architecture.

The main improvements brought by the new system design allow for a significantly higher power stability ($\sim 10^{-2}$ dB) and a considerably increased time resolution—up to 50 ms. We remark that, with respect to SWAMM-1.0, SWAMM-2.0's power stability improved by a factor of 10, while its measurement time resolution was even more enhanced by a factor greater than 20. The architecture of the previous prototype limited the measurement time resolution at more than one second because of its signal synthesizer. The synthesizer transmitted one tone at a time with a duty cycle of 50%. SWAMM-2.0 transmission module allows for a simultaneous transmission of both tones with a high degree of stability. The receiver has been enhanced with a new SDR transceiver module with greater computing power. Apart from the simultaneous transition and reception, the improvements introduced in the measurement time resolution were possible even due to the faster data processing on the receiving side.

The SWAMM-2.0 measurement campaign started in the beginning of August 2021 and lasted until the end of November 2021. The sites chosen for the TX (Transmitter) and the RX

(Receiver) were the top of Mt. Gomito (44°7′39.30″ N, 10° 38′36.81″ E) at 1900 m a.s.l and the CNR research institute in Florence (43°49′4.10 N, 11°12′1.39″ E) at 50 m a.s.l, respectively. As for SWAMM-1.0, we observed that the received SWAMM-2.0 signals were still affected by considerable power fluctuations. Even though the topographic conditions of the chosen link were less than optimal in the SWAMM-2.0 experimental campaign (the signal is more affected by multi-path effects), these fluctuations had much lower excursion levels, 3 times less in the worse case. This fact was the first to indicate that the new system's uplifts observed during the tests in the anechoic chamber were reflected in the measurement campaign's data. Given the inappropriate environment's topology combined with the horn antenna's poor directivity, we alluded that this issue persisted due to the heavy multi-path effects on the receiver, which were up to the point where the direct received signal was irreversibly impaired. To confirm our doubts, a new, more directional parabolic antenna was installed in the receiver on 23 November, consisting of enhanced gain of 38.6 dBi and narrower half power beamwidth (<2 degrees). Figure 3 shows the raw signals for both 18.8 and 19.2 GHz (blue and red curves respectively, upper panels) and the difference (green curve, lower panels) during the same time periods of two hours (from 12:00 to 14:00) at two different days with similar conditions of integrated water vapour, each with a different RX antenna; left panels (22 November) refer to horn antenna while right panels (24 November) refer to the parabolic one. The use of the more directional parabolic antenna reduces the multi-path effects and significantly reduces the signals' power fluctuations, as expected. The antenna comparison suggests, for a future campaign, the use of high directivity antennas for both the TX and RX. Furthermore, different installation sites can be considered for obtaining a more appropriate link, less affected by the multi-path effects.

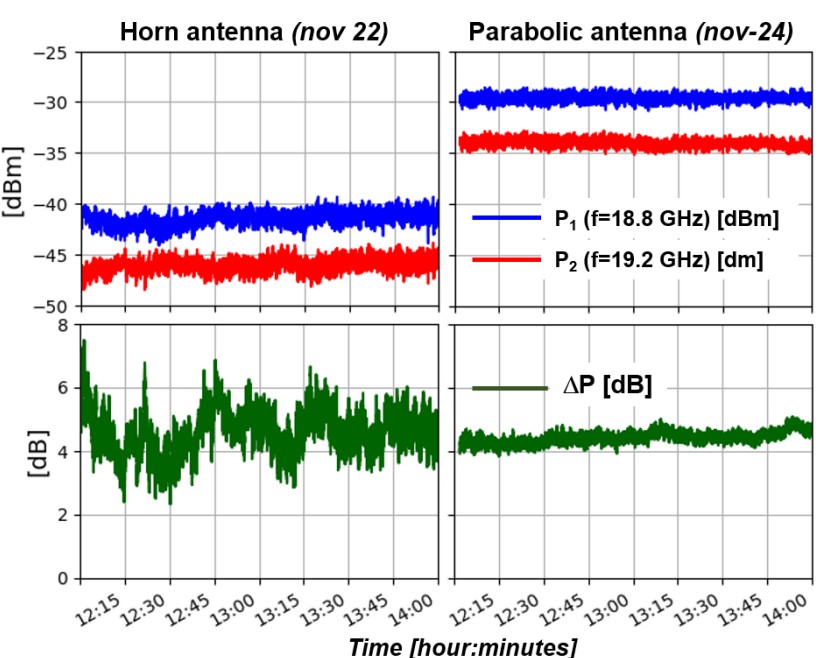

**Figure 3.** Comparison between measurements performed with different receiving antennas.

## 3. NDSA Methods for Tropospheric Water Vapour Retrievals and Applications

In this section, we summarize the inversion methods that allow the WV content to be estimated from IWV measurements and that have been proposed within the SATCROSS project and in previous studies [23,26].

Let us denote the water vapour function to be reconstructed as $f(r, \theta)$, expressed in polar coordinates. The measured IWV field coincides with its Radon Transform, indicated in the following as $g(\rho, \phi)$ and defined as

$$g(\rho, \phi) = \int_{L(\rho, \phi)} f(r, \theta) du, \tag{3}$$

where $L(\rho, \phi)$ is a line identified by the couple $(\rho, \phi)$, with $\rho$ as the distance of $L$ from the origin and $\phi$ the angle formed by the normal to the line with respect to the horizontal axis. The acquisition system geometry is depicted in Figure 4, where the radial variable has been normalised with respect to the inner circle's radius.

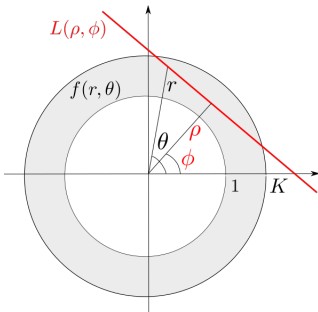

**Figure 4.** Computation of the Radon transform.

In recent studies, different approaches have been devised to solve the IWV's WV inversion problem. In [23], a least squares approach has been proposed to achieve the WV content on a discrete grid in the spatial domain: the IWV quantities are expressed as a linear combination of the unknown WV values on the grid, so that the final solution can be achieved by solving a linear system, with the addition, possibly, of a regularizer. In [26], the decomposition of the spatial function $f(r, \theta)$ and of its Radon transform $g(\rho, \phi)$ in particular Hilbert spaces have been used to solve the so-called *external reconstruction problem*. In the following, the two methods are briefly summarized.

*3.1. Least Squares Inversion Method*

Consider Figure 5, where the grid of WV variables and the line, along which an IWV measurement (a sample of the Radon transform) is taken, are depicted.

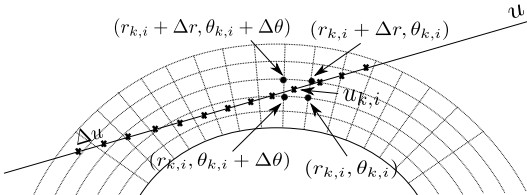

**Figure 5.** Computation of the integral WV.

Let $g_k$, $k = 1, 2, \ldots, K$, be the set of Radon samples computed by the acquisition system and let $u$ denote the spatial variable along the line used to compute $g_k$. If we approximate the integral with a discrete summation over points $u_{k,i}$ along the line, we have

$$g_k = \sum_i f(u_{k,i}) \Delta u \tag{4}$$

where $\Delta u$ is the sampling step along the line. Since, in general, $u_{k,i}$ does not belong to the sampling grid of $f$, bilinear interpolation can be used to compute $f(u_{k,i})$ from the four closest samples surrounding $u_{k,i}$, as shown in Figure 5. Hence, the quantity $g_k$ is a linear

combination of the unknown vector $\mathbf{f}$, which collects the values of the function over the spatial sampling grid, that is,

$$g_k = \mathbf{a}_k^T \mathbf{f} \tag{5}$$

where $\mathbf{a}_k$ contains the interpolation coefficients. Thus, if $\mathbf{g}$ denotes the vector collecting all the IWV measurements (sampling of the Radon transform domain), we have

$$\mathbf{g} = \mathbf{A}\mathbf{f} \tag{6}$$

where $\mathbf{A}^T = [\mathbf{a}_1 \ \mathbf{a}_2 \ \cdots \ \mathbf{a}_K]$.

A least square (LS) approximation of $\mathbf{f}$ can be achieved by

$$\hat{\mathbf{f}} = \mathbf{A}^\dagger \mathbf{g} \tag{7}$$

where $\mathbf{A}^\dagger = (\mathbf{A}^T\mathbf{A})^{-1}\mathbf{A}^T$ is the (Moore-Penrose) pseudo-inverse of $\mathbf{A}$. It is known that $\mathbf{A}$ is usually an ill-conditioned matrix and the calculation of $\mathbf{A}^\dagger$ is carried out exploiting the singular value decomposition (SVD) [23].

In the case of an overdetermined problem (number of measurements greater than the number of unknowns), the LS approximation yields the solution that minimizes the error norm:

$$E = \|\mathbf{g} - \mathbf{A}\mathbf{f}\|^2 \tag{8}$$

In the case of an underdetermined problem, a unique solution can be achieved by imposing a regularization term to the error to be minimized, that is,

$$E = \|\mathbf{g} - \mathbf{A}\mathbf{f}\|^2 + \lambda \|\mathbf{L}\mathbf{f}\|^2 \tag{9}$$

where $\mathbf{L}$ is a linear operator (for instance, a high-pass filter may be used to induce a smooth solution). Obviously, regularization can also be useful in the overdetermined case.

The solution of this problem (the Tikhonov approximation) is given by [27]

$$\hat{\mathbf{f}} = (\mathbf{A}^T\mathbf{A} + \lambda \mathbf{L}^T\mathbf{L})^\dagger \mathbf{A}^T \mathbf{g} \tag{10}$$

The advantage of such an estimator is that, with proper choices for $\mathbf{L}$ and $\lambda$, the system matrix becomes better conditioned with respect to $\mathbf{A}$ itself and the effects of disturbances and numerical errors can be treated more effectively.

### 3.2. Exterior Reconstruction Method

An alternative approach of tomographic inversion, perfectly coherent with the geometry of the inversion problem at hand, has been proposed in [26]. Its original formulation, by Cormack, dates back to the 1960s [28,29] and has been revisited and analysed in the following decades by Cormack himself [30], Perry [31], and Quinto [32–34], with applications to radiology and non-destructive testing. The mathematical problem posed in these works is inverting the Radon transform of a 2D field calculated on a compact annular region, which perfectly fits the scenario taken into consideration in our study. Since the functions $f(r, \theta)$ and $g(\rho, \phi)$ are periodic in the angle variables $\theta$ and $\phi$, they can be expanded as Fourier series; that is,

$$f(r, \theta) = \sum_{n=-\infty}^{\infty} f_n(r) e^{jn\theta}$$
$$g(\rho, \phi) = \sum_{n=-\infty}^{\infty} g_n(\rho) e^{jn\phi} \tag{11}$$

A set of important theoretical results has been demonstrated in the works by Cormack, Perry, and Quinto. First, it can be shown that each term of the two series can be further decomposed by means of specific bases of functions, namely $f_{n,l}(r, \theta)$ and $g_{n,l}(\rho, \phi)$, $l \in \mathbb{N}$, whose major property is that the Radon transform maps such functions onto each other [31].

Such elementary functions are expressed by means of Jacobi polynomials (their definition is not given here for the sake of conciseness). It can also be shown that there exist non-null functions $f_{n,l}(r, \theta)$ that are mapped onto zero by the Radon transform: such functions generate the *null space* of the transform. This fact implies that, since in our problem the observable quantities are samples of the Radon transform, a non-null component belonging to the null space cannot be reconstructed from the measurements: thus, such a component cannot be inferred from observations and may have a role in the reconstruction of the function $f(r, \theta)$ only based on some a priori knowledge (for example, if we knew that the solution is smooth). Another important fact is that the elements $f_{n,l}(r, \theta)$ and $g_{n,l}(\rho, \phi)$ are orthogonal in the space domain and in the Radon domain, respectively, if particular weight functions are used to define the inner product in such spaces. Furthermore, they form the basis of the respective spaces.

Therefore, taking into consideration all these properties, a procedure to reconstruct the WV field from the IWV measurements can be devised. The basic steps are the following: (1) decompose the sampled Radon transform $g(\rho, \phi)$ (IWV) by using the set of elementary functions $g_{n,l}(\rho, \phi)$; (2) use the coefficient of the decomposition and the elementary functions $f_{n,l}(r, \theta)$ to synthesize $f(r, \theta)$ (WV).

### 3.3. Applications

Examples of the application of the inversion methods described above are now presented. As for the ground truth, we use the atmospheric scenario shown in Figure 6, which is relative to the entire orbital plane drawn from the ECMWF, T511 L91 analysis data of 15 January 2011, 00:00UTC (used also in [23]). The vertical and angular sampling intervals are 125 m and 1 degree, respectively; the WV concentration field is in polar format, but it is actually shown by means of a rectangular grid, with the *x*-axis and *y*-axis referring to the angular position and the altitude, respectively. For our WV retrieval testing purposes, we did not consider the liquid water content present in such a scenario.

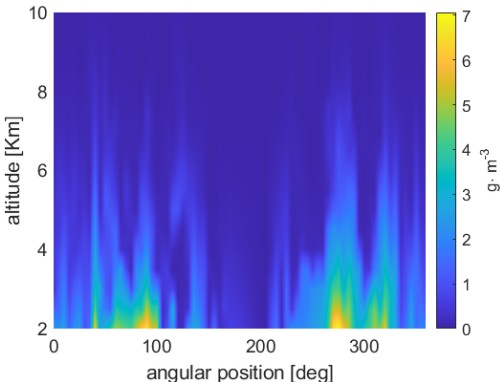

**Figure 6.** Original atmospheric WV map. The field (in polar format) represents an entire annular region, corresponding, approximately, to the orbital plane during a revolution of the satellites. It is shown here as a rectangular grid, in which the *x*-axis and the *y*-axis refer to the angular position and to the altitude, respectively.

Figure 7a,b show the reconstructions that were achieved with the algorithms described in Sections 3.1 and 3.2, respectively. The following assumptions about the acquisition system and satellite constellation geometry were made:

- Ideal transmission and acquisition conditions (absence of thermal noise and scintillation effects);
- One Tx satellite and five Rx satellites on the same circular orbit;
- Polar orbit with a satellite altitude equal to 273 km;
- Constant angular speed with a revolution period of 90 min;
- Integration time at the receiver $T_s = 1$ s.

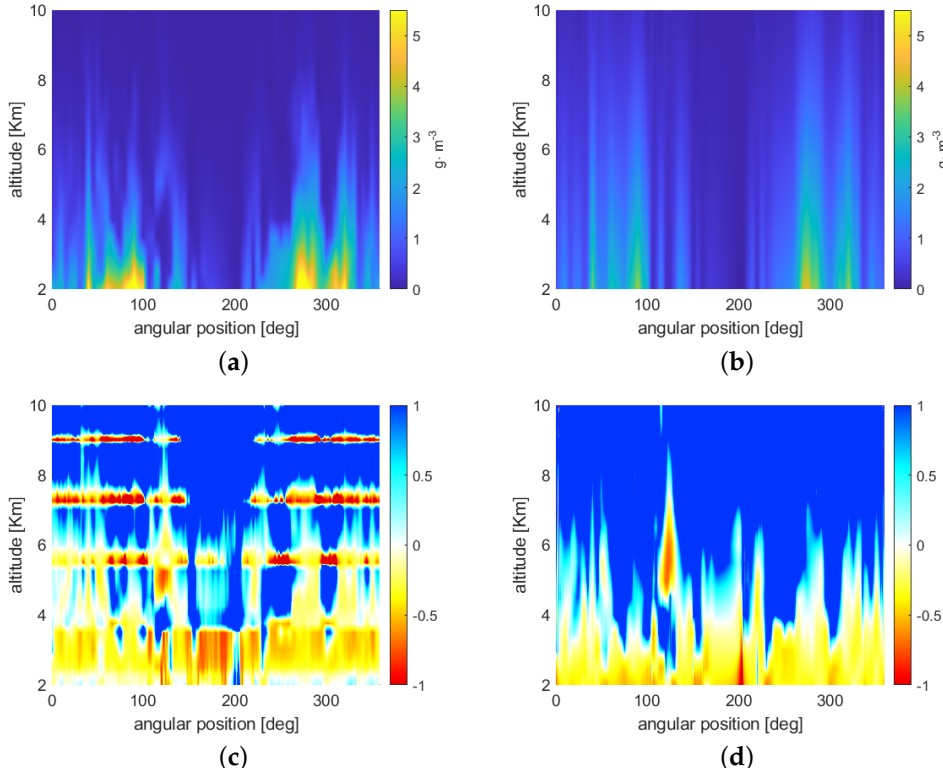

**Figure 7.** Reconstructed WV maps obtained with the least squares algorithm with Tikhonov regularization (**a**) and with the exterior reconstruction solution (**b**). In plots (**c**) and (**d**) (for the same methods and in the same order), the errors between the retrieved WV field and the original one, normalised with respect to the latter quantity, are shown.

In order to better appreciate the reconstruction quality, we show in Figure 7c,d the error between the retrieved WV field and the original one, normalised with respect to the latter quantity.

Observing the retrieval results, it is apparent that the two methods produce reconstructions and error maps with different spatial behaviour. In fact, the least squares approach is characterized by 'bumps' in correspondence of the tangent altitudes of the satellite links, whereas the exterior reconstruction method exhibits a spatially regular trend. The reason for this behaviour lies in the quadratic minimization at the base of the least squares method, which tends to force to zero the error in the measurement points (the tangent altitudes); this effect is not present in the exterior reconstruction case that is based on the combination of relatively regular components. In general, at the lowest altitudes the reconstruction error produced using the latter approach are lower than those produced by the former one, while, above a certain altitude, depending on the WV field, the exterior reconstruction constantly overestimates. This may again be ascribed to the fact that the basis functions used to represent the Radon transform and, successively, to reconstruct the WV field are quite regular, and, in general, exhibit a decay that is slower than the WV field variations.

## 4. Other Related Considerations for NDSA Retrieval and Space Deployment

### 4.1. Correction of Tropospheric Liquid Water Contents on WV Retrievals

The IWV estimates provided by the NDSA method are affected by the presence of liquid water (LW) along the radio link. Specifically, the impact of LW on the IWV estimates provided by NDSA in the 17–21 GHz frequency range is a positive bias, which is proportional to the total integrated liquid water (ILW) content along the radio link. The effects of the presence of ILW were analysed in [21,35], where the use of the spectral sensitivity measured in the 30–32 GHz range was also proposed for estimating the path of integrated LW, and possibly for correcting the IWV measurements. However, such an

approach based on the spectral sensitivity comes out to be quite limited even when the signal-to-noise ratio of the received signals is relatively high. On the other hand, it is possible to estimate the ILW through a different approach, i.e., by means of the power ratio of the received signals at 31.8 GHz and at 16.9 GHz.

Let us define $P_{17}$ and $P_{32}$ as the received powers at 16.9 and 31.8 GHz, respectively, and $R_{17/32}$ as follows:

$$R_{17/32} = -10log\frac{P_{32}}{P_{17}} \tag{12}$$

And let us define the spectral sensitivity $S_{32}$ as:

$$S_{32} = 1 - \frac{P_{32.2}}{P_{31.8}} \tag{13}$$

where $P_{32.2}$ and $P_{31.8}$ are the received powers at 32.2 and 31.8 GHz, respectively. Using an end-to-end simulation tool for simulating the signals on the receiving satellites, we simulated the $S_{32}$ and $R_{17/32}$ parameters, assuming some reference atmospheric scenarios.

The reference atmospheric model used for simulating the ILW effects refers to 24 July 2020. That day was characterized by an advancing warm front, capable of generating convection with a limited vertical speed, and by the rapid passage of a precipitating event on the central and northern parts of the Italian peninsula. The annular sector considered for the co-rotating propagation simulations reported in this paper is the [34°–46°] latitude North at 11.15° longitude East, with an altitude between 2 and 10 km. The vertical resolution is 0.125 km, and the latitude resolution is 3 km. Concerning the acquisition system and the satellite constellation geometry, the following parameters have been set:

- Satellites co-orbiting on a circular polar orbit at constant angular speed, having a revolution period of approximately 90 min;
- Earth radius: 6378 km;
- Satellite orbit radius: 6651 km (273 km altitude);
- Integration time at the receiver: Ts = 0.5 s;
- Transmitted power: 3 dBW for each tone;
- Frequencies for power measurement: 16.9, 17.1, 18.9, 19.1, 20.9, 21.1, 31.8, and 32.2 GHz;
- Tx and Rx antenna gains: 26.4 dB;
- System noise equivalent temperature: 25.3 dBK.

Path losses due to atmospheric absorption, free space, and defocusing have been also considered. We consider a constellation geometry like that sketched in Figure 1 (one transmitting and three receiving satellites on the same circular orbit), where the satellites are placed so that the tangent altitudes (which remain constant in time assuming circular orbit and spherical Earth) of the three radio links are 2, 5, and 9 km. We also assume that the initial and final positions of the satellites are such that the Tx-Rx link at the 2 km tangent altitude intersects the points (34°, 10 km) and (46°, 10 km), respectively. In this manner, all the radio links cross the considered annular sector spanning [34°–46°] in latitude and [2–10] km in altitude, and the number of time samples for each Tx-Rx link is 158.

Figures 8 and 9 show the time sequences of $P_{17}$ and $P_{32}$, respectively, and Figure 10 shows the time sequence of $R_{17/32}$ under the assumptions made above. Notice that: (1) the $R_{17/32}$ trend follows that of the ILW at all the three tangent altitudes and (2) the power level of both $P_{17}$ and $P_{32}$ is greater than the noise power level, which is about $-200$ dBW for the selected Tx-Rx setup.

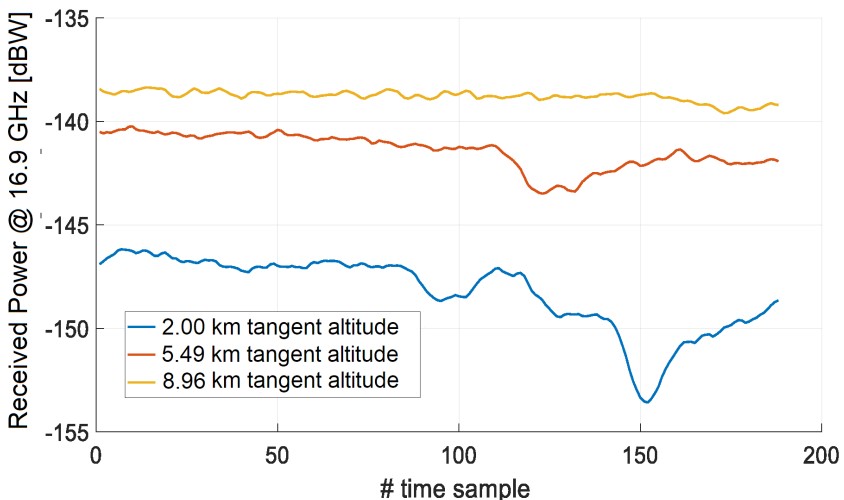

**Figure 8.** Time sequence of the received power at 16.9 GHz.

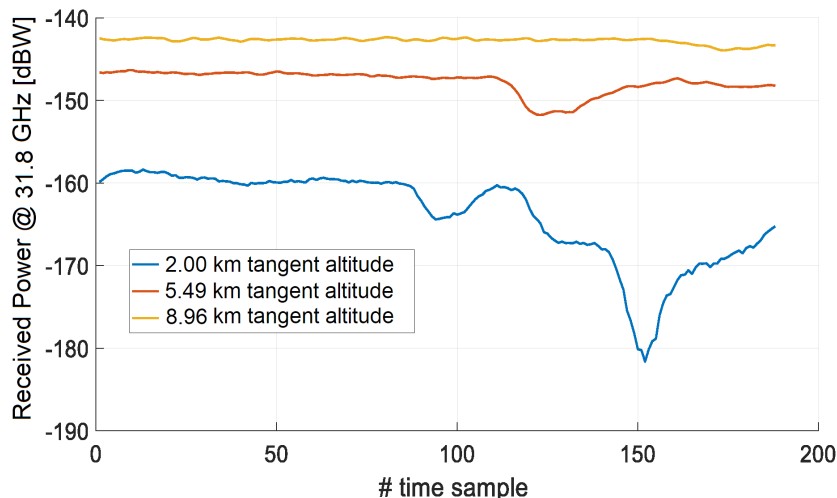

**Figure 9.** Time sequence of the received power at 31.8 GHz.

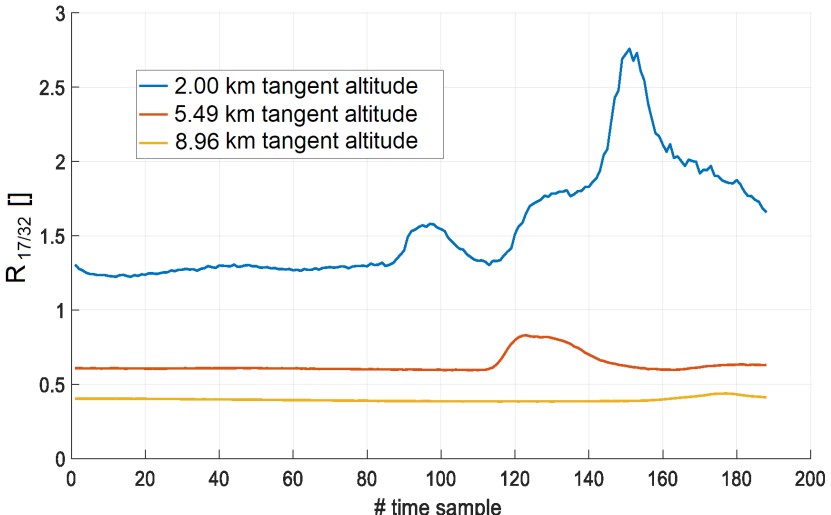

**Figure 10.** Time sequence of the *R*17/32 parameter.

Figure 11 shows the scatter plots between the ILW and $S_{32}$ for all tangent altitudes in both ideal (no impairments) and realistic propagation conditions. Notice that the scatter of the 'real' measurements of $S_{32}$ with respect to the ideal ones is excessive, which does not allow one to provide a direct estimate of the ILW with acceptable precision, even though the relationship between ILW and $S_{32}$ in ideal conditions is actually that reported in [21].

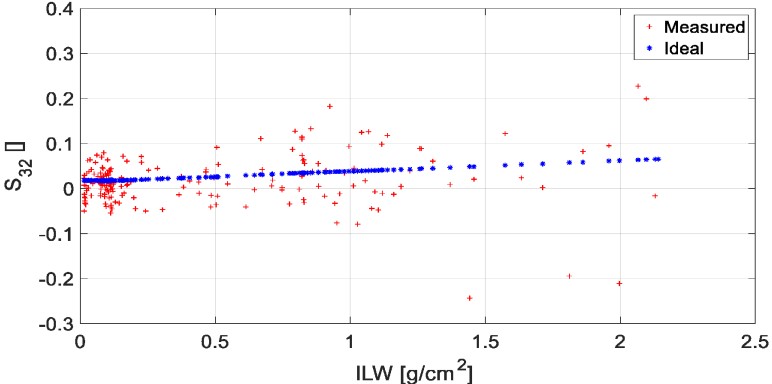

**Figure 11.** Scatter plots between ILW and $S_{32}$ for all tangent altitudes in ideal (no impairments) and true propagation conditions.

Figure 12 shows the scatter plots between ILW and $R_{17/32}$ separately for the three tangent altitudes. In this case, a linear trend is evident, and this suggests that the measurement of $R_{17/32}$, even in realistic conditions (i.e., accounting for impairments), could provide a direct estimate of the ILW via a linear conversion law. However, such a possibility needs to be confirmed by testing several different atmospheric conditions in addition to those used in this study.

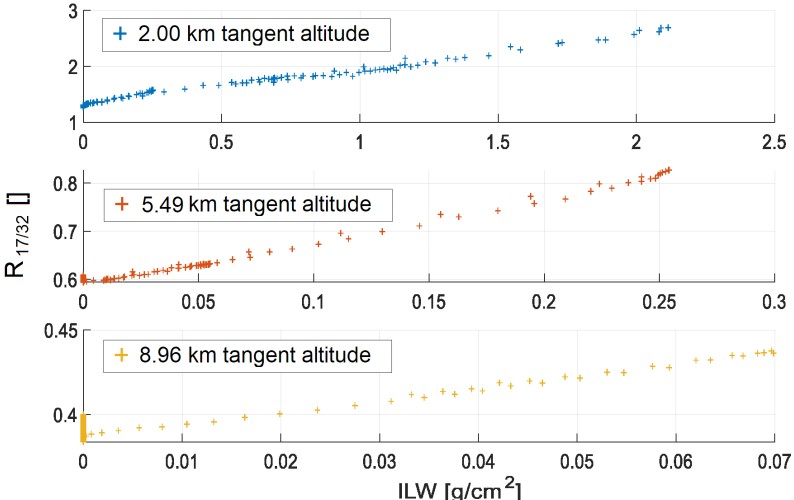

**Figure 12.** Scatter plots between ILW and $R_{17/32}$ for the three tangent altitudes.

As a matter of fact, the conversion factor between $R_{17/32}$ and ILW needs to be computed by linear regression while considering as many different atmospheric scenarios as possible to correctly account for the global variability, adopting, for instance, the approach followed in [17]. Finally, note that the presence of even a moderate amount of LW for an extended part of the radio path may cause a significant attenuation of the signal power. ILW values can be estimated as long as the received power of the 32 GHz signal exceeds the noise power level. The upper limit of the ILW that can be estimated has a greater impact on the estimates made along the radio links at the lowest tangent altitudes.

*4.2. Mission and Payload Concept Demonstration*

SATCROSS will be realised more easily and appropriately if it is implemented through a CubeSat-based mission. To this end, the space segment requires to be defined with respect to scenario and payload design.

Mission scenario analysis has been developed to primarily set the orbital parameters, constellation architecture, and launch configuration. First of all, the orbital altitude is constrained by a set of conditions. The link budget requires a shorter distance between the transmitting and the receiving satellites: this means a reduced altitude is preferred, possibly one not exceeding 400–500 km. Similarly, ionising radiation effects exclude altitudes too far above 400 km: in that case, radiation shielding protection would overload the CubeSat payload. On the other hand, atmospheric drag causes too short of a mission duration in cases where the orbital altitude is far below 400 km. Therefore, finally, the proper quote results in about 400 km. As a further side remark, this orbital altitude matches the International Space Station (ISS) one: a viable possibility may consist in accommodating the scientific transmitter on board the ISS, so that only the receiving satellites would have to be properly arranged.

Software simulations provide additional information on orbit and constellation design. The simulation's main goal has consisted of minimising the revisit time (namely, the time elapsed between two consecutive satellite crossings) over a mid-latitude target area, while guaranteeing an even coverage, i.e., lowering the spread of the values. With reference to orbital inclination, a sun-synchronous orbit both guarantees the coverage of all latitudes and allows for a longer average coverage time, when compared with other inclination values. So, this solution presents the benefit of being versatile since it can provide similar performances independently from the region of interest. On the opposite, when the target is restricted to just a specific area, it is possible to reduce the revisit time by lowering the inclination value according to the location latitude; however, this comes with the drawback of ruling out from coverage all the regions whose latitude overcomes the orbit inclination value.

Regarding the constellation architecture, different configurations have been investigated using simulations. For this purpose, a set of one transmitting and three receiving satellites has been defined as the basic unit, also called a train of satellites. So, simulations have involved up to four trains of satellites distributed along one, two, and four orbits (denoted as 4s_ 1p, 4s_ 2p, and 4s_ 4p, respectively). In each of these three cases, the satellites' true anomaly and right ascension of the ascending node (RAAN) values have been set so that the satellites are evenly distributed along each orbit and around the Earth. Figure 13 helps in understanding the revisit time behaviour over a simulated month. Therein, oscillations result as the periodic transit of the orbits above the region of interest due to the Earth's rotation. Peaks in the revisit time values occur every time the target region rotates away from under a constellation orbit. Low revisit time values are related to the condition when the region of interest is underneath an orbit plane. The revisit time values are linked to the number of planes available in the constellation, the number of satellite per plane and to the orbital period. The simulation results showed that, for sun-synchronous orbits, the revisit time distribution is more uniform when the ratio between the total number of trains of satellites and the number of orbit planes is close to 2:1. So, using more than two planes does not improve the performance, because it reduces the coverage time and the satellite crossing probability. In conclusion, the best result for general Earth coverage consists of a constellation of four trains of satellites evenly spread over two sun-synchronous orbits, 90° (RAAN) apart from each other.

Smallsat platforms, as CubeSats are, force payload design to undergo a significant miniaturisation. So, the size, mass, and power consumption must be limited to the CubeSat assets: a 12–16 U one is attainable in the present case. On one hand, satellite systems that are compatible with such requirements are already available on the market; what is more, they are provided with technology readiness level (TRL) equal to 9. On the other hand,

the specific scientific instrumentation needs a new dedicated development, although it has already achieved TRL 3–4.

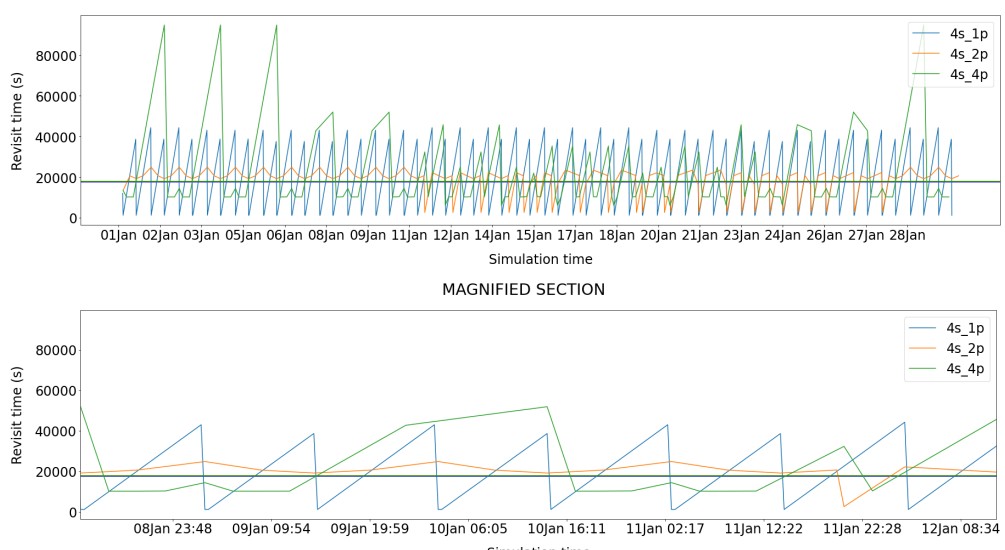

**Figure 13.** Simulated revisit times over an atmospheric region above mid-latitudes: 1-month simulation (above); zoom covering a few days (below).

The purpose-built scientific equipment consists of an antenna and either a transmitter or a receiver. Regarding the scientific antenna, gain is recommended starting from 30 dB. For this aim, corrugated horn or axially displaced ellipse or reflector antennas are the preferred candidates: in particular, the latter has recently been receiving major interest from CubeSat-telecommunication providers, achieving a gain greater than 40 dB. The scientific transmitter and receiver preliminary design can be facilitated by using commercial off-the-shelf (COTS) components, which, moreover, reduce production costs and times. Solutions based on either analogue (PLL, RSSI) or digital (FPGA) architectures should be taken into account for both devices. The scientific transmitter case may be nimbly solved by an analogue architecture, where two high-frequency tones are synthesised separately, to be subsequently combined at the front-end input. Here, a robust power amplification chain must allow the final RF signal to abundantly exceed power levels above 30 dBm, at the same time paying a great deal of attention to the intermodulation of the products. On the other hand, the selectivity of the scientific receiver must be accurately elaborated, since the two very close signal tones have to be acquired together and detected separately. This can be translated into a double down-conversion super-heterodyne. The down-conversion would be combined with frequency tone selection, while the second conversion would be devoted to signal selectivity. This can be achieved using either PLLs or FPGAs; the latter digital solution may be preferred in order to dynamically vary selectivity, and to start up the first data processing at the raw data level.

### 4.3. Assimilation Impact Assessment of WV Products from NDSA Measurements

In order to evaluate the impact of NDSA measurements on environmental model simulations, an Observing System Simulation Experiment (OSSE), whose steps are summarised in the scheme of Figure 14, was used as a reference approach.

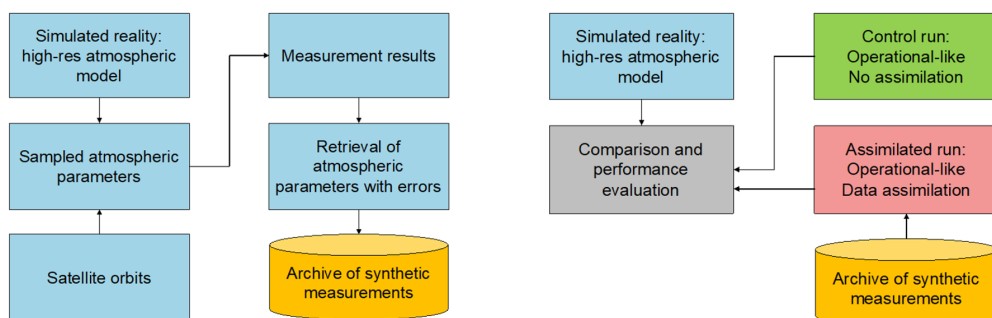

**Figure 14.** The Observing System Simulation Experiment (OSSE) approach for generating synthetic measurements (phase 1 in left panel) and for the analysis of the assimilation impact on the model forecasts (phase 2 in right panel).

Typically, the OSSE develops through two phases: phase 1 for generating synthetic measurements from some realistic simulated scenarios, called virtual reality (VR), and phase 2 for estimating the impact that the (synthetic) measures, derived from phase 1, may have on the forecasts, through appropriate assimilation procedures. Phase 2's evaluation is achieved by comparing any simulation performed with data assimilation (ASSIM) with the control simulation, performed in the same way but without data assimilation (CTRL), against the corresponding VR simulation they should look like. In the first phase, the reference scenarios are usually generated with a 'better' simulation model that performs predictions closer to nature to be imitated with respect to that used in the second phase. This can be achieved through a higher resolution or better initial/boundary conditions, better physics, etc., or a combination of these factors. Synthetic measurements are generated from this virtual (or synthetic) reality, based on the sampling capacity of the instrument under study and using the appropriate recovery procedures. These measurements are stored and used in phase 2 for assimilation on a simulation cycle, typically imitating an operational forecast. In general, different initial and boundary conditions and a different resolution determine different paths in the numerical models, and therefore, even though it is not a linear process, the effect of assimilation should be to force the forecast towards the virtual reality. It should be noted that the assimilation paths to be tested may possibly be different, depending on possible different assimilation techniques and/or different measurement characteristics (number, distribution, uncertainty, type, etc.).

In both OSSE phases, we used the same meteorological model, namely the Weather Research and Forecasting model (WRF) [36]. It is a NWP system developed for both atmospheric research and operational forecasting objectives. It is a fully compressible, Eulerian, non-hydrostatic mesoscale model, and we implemented the Advanced Research WRF (ARW) version 4.1. A description of the model dynamics, equations, and numerical schemes can be found in [36,37], while the model physics are in [38]. We used the same implementation for both phases and the differences were obtained with only different initial and boundary conditions. The geographical domain included the whole Italian peninsula.

Three different atmospheric scenarios were generated and used for assimilation purposes. The latter are representative of meteorological conditions with different levels of severity with respect to air mass motion and precipitation. For the sake of synthesis, we detail the main qualitative and quantitative results only for the more meteorologically complex case study, i.e., 24 September 2020, in which water vapour and liquid water contents appear in large quantities. The modelling architecture is described in detail in Appendix A. The assimilation was carried out in a window of ±15 min centered at 18UTC, by providing as inputs to the model the values of the PW and WV vertical profiles obtained from the NDSA-retrieved process relative to the transept over Italy (Figure 15, dashed line in the red box of the top left panel). The comparison between VR, CTRL, and ASSIM was carried out in the range between 18UTC of 24 September and 00UTC of 25 September, checking the accumulated rainfall for the following 6 h.

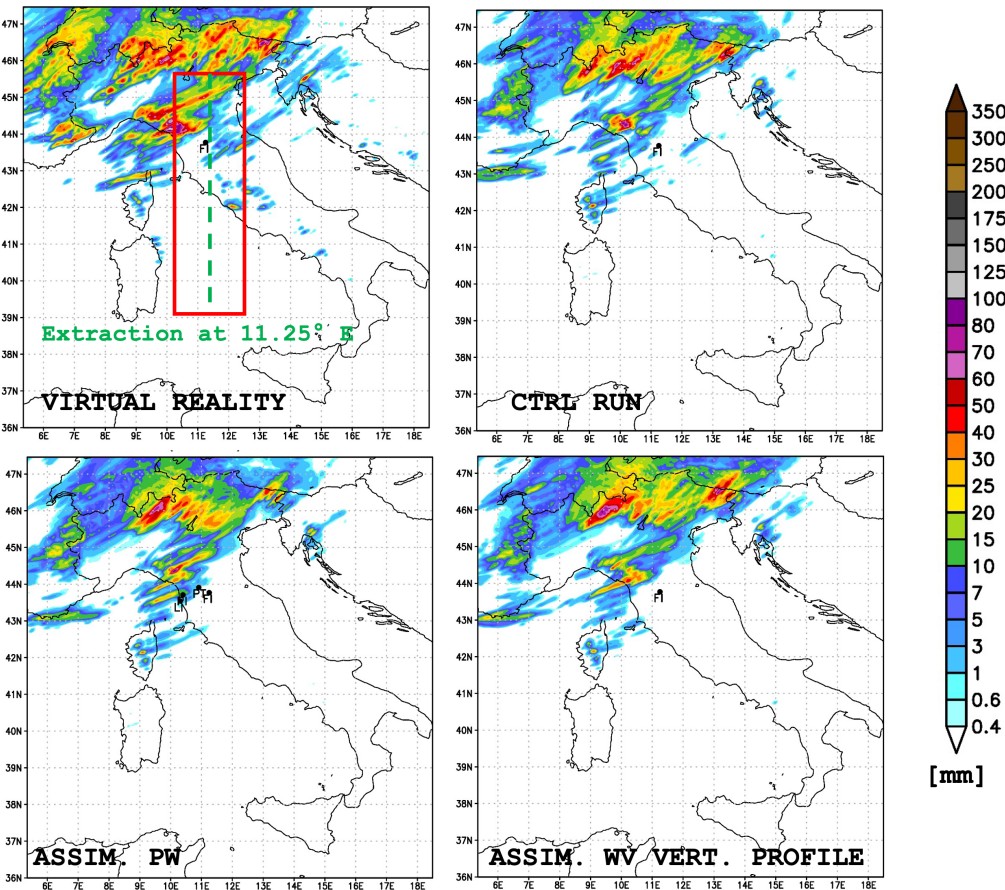

**Figure 15.** Case study of 24 September 2020: 6-h accumulated precipitation for 18-00UTC. VR (top left), CTRL (top right), ASSIM forecasts with assimilation of precipitable water values (bottom left) and WV vertical profiles (bottom right). Note the impact of the data assimilation in a better characterization of the precipitative event approaching the Tuscan coast (compared to CTRL).

Notwithstanding the low differences between CTRL and VR and the present magnitude of the NDSA measurement errors, ASSIM shows a clear impact of the NDSA data assimilation, even if, we remind, it is done just once in the forecast simulation (i.e., at one single assimilation time) and just for a single transept in the simulation domain. It is in fact noted that the PW assimilation and the assimilation of the WV profile, on the selected transept positions, make the ASSIM run reproduce the precipitating systems approaching the Tuscan coast and extending towards Emilia-Romagna more accurately, with respect to the CTRL run (Figure 15). This correction is positive as it tends to bring the model dynamics closer to VR and it has relevant effects on the evolution of the forecast. In the upper part of the domain, ASSIM substantially confirms what was obtained with CTRL, and, especially in the western side of the domain, provides some improvements in the estimation of rainfall intensities, bringing the rainfall patterns closer to those of VR.

Of course, the impact of the assimilation is also spread aside from the transept by the assimilation technique and over other meteorological variables than the assimilated one (e.g., wind speed and direction, temperature, and relative humidity). Some quantitative results for these variables through a statistical analysis are detailed in Appendix A (see Tables A1–A3).

## 5. Conclusions

The NDSA technique is supported by a strong theoretical background, which demonstrated its ability to provide accurate measurements of the total water vapour content (the IWV) along a microwave link. In this paper, we have outlined the main results of the

SATCROSS project, which concerns the application of the NDSA technique to the case of a train of LEO satellites rotating in the same orbital plane and providing sets of IWV measurements from which one can retrieve 2D water vapour fields on planes perpendicular to the Earth's surface. The results of the SATCROSS project constitute a solid basis for progress towards a space mission for the observation of the Earth's atmosphere, characterized by a very original approach. The inversion algorithms are evidently crucial for retrieving the aforementioned 2D fields from sequences of separate IWV measurements referring to links between pairs of LEO satellites. Two algorithms have been developed and tested, showing that the inversion can be achieved by resorting to completely different approaches. As a consequence, the algorithms outputs (the retrieved fields) also exhibit different features, even if their average retrieval errors do not differ much from each other. The possibility of detecting the presence of LW along each microwave link and correcting the WV estimates made along paths where LW is detected has been revised, demonstrating that the ratio of the received powers at 32 GHz and 17 GHz is more appropriate for this purpose than the use of an additional NDSA channel providing the spectral sensitivity at 32 GHz.

The other pillars of the SATCROSS project were the definition of reference atmospheric scenarios and the development of an end-to-end tool simulating the received signals and disturbances. Besides being exploited as input for developing and testing the end-to-end simulator and inversion algorithms, the simulated atmospheric scenarios were also of fundamental importance to the subsequent evaluation of the impact on the forecasting modelling chains. In this regard, the impact of the NDSA products has been evaluated through an OSSE (Observation System Simulation Experiment) approach. The performed numerical experiments were limited in terms of observation distributions and errors, assimilation methods, types of weather events, etc. Therefore, definitive conclusions cannot be drawn, even if the results (reported only in part in this work) clearly point out the relevance of the NDSA measurements when used in an atmospheric data assimilation process. The moderate impact is on the precipitation patterns and their intensities, but other analysed meteorological variables are also impacted: the temperature, relative humidity, and wind are significantly affected. At the same time, the sensitivity to the final accuracy of the retrievals and the need for further improvements emerge. Different case studies and assimilation approaches are still under evaluation and further experiments are underway. Their objective is to improve the OSSE capability to provide sounder answers on the effective potential of assimilating NDSA measures for increasing future operational forecast skills.

The study pointed out that a SATCROSS space mission based on CubeSat technology is feasible. The mission analysis evaluated a sun-synchronous orbit and an altitude of around 400 km as being appropriate: this choice makes a satisfactory mission duration and radio link possible and opens the door to the use of payloads based on non-radiation-hardened COTS components. Different orbital configurations were also investigated, including up to four observational units arranged on as many different orbital planes (but with the same inclination and altitude). The conclusion is that particular attention must be paid to the choice of the final architecture of the mission, depending on whether one intends to focus the observation on a specific area—possibly with very short revisit times in a certain amount of time alternating with longer revisit times—or whether a more homogeneous distribution around the planet is preferred. Given the coincidence of the altitude, the study also highlighted the importance of considering the International Space Station (ISS) option, both as a platform for one of the payloads, and as a basis for the release of a dedicated CubeSat SATCROSS into orbit.

A crucial step towards space deployment is related to the NDSA measurement prototype, which has been greatly improved in its fundamental components and tested during an experimental campaign, and will be further employed in validation campaigns where other independent measurement systems will be employed. The architectural simplification and technical upgrade of the prototype can be considered as preparatory for the subsequent strategic development and integration steps of the modules, which are necessary for the system to probe the vertical profile of water vapour in the troposphere.

**Author Contributions:** Conceptualization, A.O.; Methodology, S.M, A.O., L.R. and L.S.; Investigation, S.D.B., F.D., M.G. and L.S.; Data Curation, A.F. and L.R.; Software, A.A., F.B., F.C., A.F., M.G., A.M., F.M. and L.S.; Resources, A.A., F.B., M.G. and F.M.; Validation, A.A., F.M. and L.R.; Project Administration, L.F.; Supervision, L.F., U.C., A.G. and T.S.; Writing-original draft, L.F., F.A., U.C., F.C., F.D., A.F., S.M., A.O. and L.S.; Writing review and editing, L.F., F.A., F.B., U.C., F.C., F.D., G.M., S.M. and A.O.; Corresponding author, U.C. All authors have read and agreed to the published version of the manuscript.

**Funding:** The new findings and insights o the NDSA measurement technique resulting from the SATCROSS (SATelliti CoROtanti per la Stima del vapore acqueo in tropoSfera) research project were funded by Agenzia Spaziale Italiana (ASI) contract number 2020-2-U.0.

**Institutional Review Board Statement:** Not applicable.

**Informed Consent Statement:** Not applicable.

**Data Availability Statement:** Data is available upon request.

**Conflicts of Interest:** The authors declare no conflict of interest.

## Abbreviations

The following abbreviations are used in this manuscript:

| | |
|---|---|
| ACE+ | Atmosphere and Climate Explorer |
| ACTLIMB | Active limb sounding of planetary atmospheres |
| ALMETLEO | Alternative Measurements Techniques for LEO-LEO Radio Occultation |
| ANISAP | Analysis of NDSA technique for Inter-Satellite Atmospheric Profiling |
| ARW | Advanced Research WRF |
| ASI | Agenzia Spaziale Italiana |
| ASSIM | Assimilation |
| COTS | Commercial Off-The-Shelf |
| CTRL | Control |
| ESA | European Space Agency |
| FPGA | Field Programmable Gate Array |
| GNSS | Global Navigation Satellite systems |
| IFS | Integrated Forecasting System |
| ILW | Integrated Liquid Water |
| IWV | Integrated Water Vapour |
| LEO | Low Earth Orbit |
| LW | Liquid Water |
| NDSA | Normalised Differential Spectral Attenuation |
| NWP | Numerical Weather Prediction |
| OSSE | Observing System Simulation Experiment |
| PLL | Phase-Locked Loop |
| RAAN | Right Ascension of the Ascending Node |
| RO | Radio Occultation |
| RSSI | Receiver Signal Strength Indication |
| SATCROSS | SATelliti CoROtanti per la Stima del vapore acqueo in tropoSfera |
| SDR | Software-Defined Radio |
| SWAMM | Sounding Water Vapour by Attenuation Microwave Measurements |
| TA | Tangent Altitude |
| VR | Virtual Reality |
| WRF | Weather Research and Forecasting Model |
| WV | Water Vapour |

## Appendix A

The modelling architecture (Figure A1) for the more complex meteorological configuration envisaged VR and CTRL were both initialized at 00 UTC of 24 September 2020, and were carried out for the following 24 h. The generation of VR was made by initializing the WRF-ARW model with the global ECMWF IFS model with a spatial resolution of 0.12° over

15 pressure levels. The model outputs were generated every 5 min for the following 24 h at a horizontal resolution of 0.03° (≈3 km) over 50 model levels to allow for the extraction of parameters that are temporally and spatially consistent with the needs of the retrieval algorithm. The forecast, initialized with the NCEP-GFS model (0.25° and 35 pressure levels), was carried out for the following 24 h for all the case studies at a horizontal resolution of 0.03° and on 50 model levels. The ASSIM runs concern precipitable water (i.e., column-integrated WV) and WV vertical profile values above 2000 m, according to the measurement's characteristics under study, on a transept centered at 11.25° E. This choice is compatible with the orography of the analysis domain (apart from a few exceptions) and with the measurement geometry. The simulations were generally run for a few hours after the precipitative event and were targeted for the retrieval, assimilation, and comparison processes. A Twice Digital Filter Initialization (TDFI, [39]) was used with a time step of 12 s for all the model runs. The CV7 covariance matrix with 1 year of model runs was used as the background error, following the NMC (National Meteorological Center) technique. As stated in Section 4.3, the assimilation was carried out in a window of ±15 min centered at 18UTC, by providing as input to the model the values of PW and WV vertical profiles obtained from the NDSA-retrieved process relative to the transept over Italy (Figure 15, up left figure). The comparison between VR, CTRL, and ASSIM was carried out in the range between 18UTC of 24 September and 00UTC of 25 September, checking the accumulated rainfall for the following 6 h.

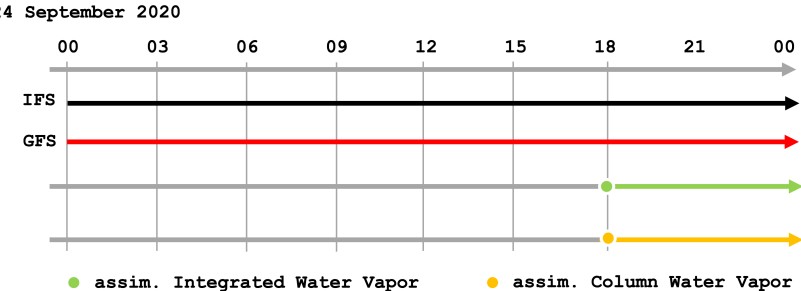

**Figure A1.** Scheme of the OSSE experiment for the case study of 24 September 2000. In black, the Virtual Reality (VR) initialized with ECMWF-IFS data; in red, the control run (CTRL) initialized with NCEP-GFS data; in green and yellow, two ASSIM simulations made at 18 UTC with the same forecast, but forced with the 3D-VAR assimilation of precipitable water and WV vertical profile data, respectively.

Figure A2 shows the values of PW extracted where the measurements transect at 11.25 E above 2000 m for the VR (blue line) and CTRL (red line) and retrieved by the NDSA technique (green line). We can see that the differences between the VR and CTRL are relatively small, always being within about 20%, apart from some points around 45° N of the latitude, where they are much higher. This generally marginal difference, specifically at the assimilation time, appeared in all the domains and for all case studies (and for different parameters), and we can say that it is too optimistic (i.e., non-realistic) with respect to the differences that we can find between a simulation and the reality of nature, at least during some critical events when forecasts did not perform as expected, and we would like to rely on data assimilation to improve the forecast skill.

This originates from the relative similarity of the initial conditions (IFS and GFS), which have several commonalities, including sharing a large part of the global observations they start from, as well as the proximity of the assimilation time to the start time of the simulations, that do not allow them to develop larger differences. In turn, if we exclude the points around 45° N, the NDSA measurements retrieved with the present technique show errors similar to or higher than this difference (e.g., where the plain mixed with mountains prevails), and this limits their potential effectiveness in improving the ASSIM forecast, that, not surprisingly, can be even worsened.

We can observe a difficulty in the reconstruction of the WV patterns, especially where we find its highest variability (both horizontally and vertically), sometimes too roughly smoothed in the retrieval. The greatest differences in terms of the PW, with respect to the chosen transept, occur beyond 43° N (apart from the few points around 45° N) when the orography influences the physical variable under study. (We remind also that a critical source of measurement errors can be the presence of liquid water that it is not explicitly managed by the present retrieval method).

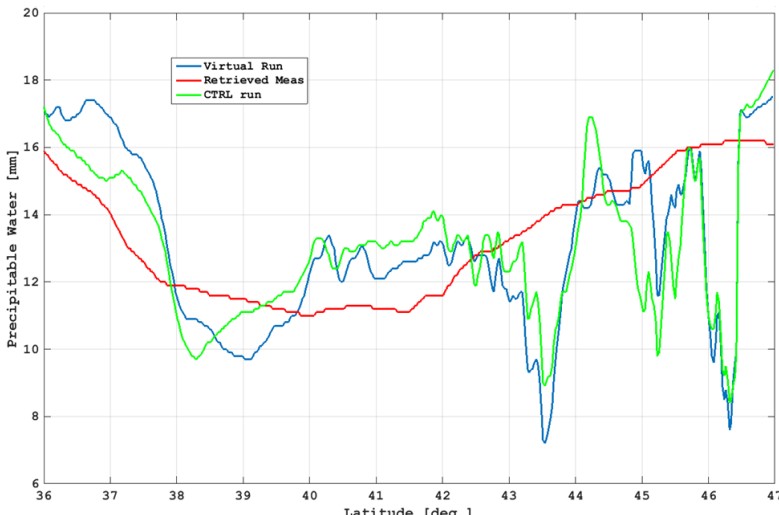

**Figure A2.** Case study of 24 September 2020, 18UTC: precipitable water values in [mm] as extracted on the transect at 11.25° E for VR (blue line), CTRL (green line), and retrieved measurements (red line).

For validation purposes, a statistical analysis was performed in approximately homogeneous areas by orography, in order to extract some interpretable information from the assimilation of NDSA measurements about the reconstruction of the atmospheric fields. Considering the complexity of the Italian territory, three areas were selected: a pure sea region (named Sea), a mix of valleys and Apennine reliefs (named Valley), and a high mountainous region on the Alps (named Mountain) (Figure A3).

In Tables A1–A3, the results of the statistical analysis are shown in terms of BIAS and RMSE for the temperature at 2 m (T2m), relative humidity (RH), and accumulated precipitation for the three areas in Figure A3. The statistics have been computed up to 6 h after assimilation and the results are reported as the differences between the two ASSIM runs (of both PW and WV vertical profiles) and VR, and CTRL and VR.

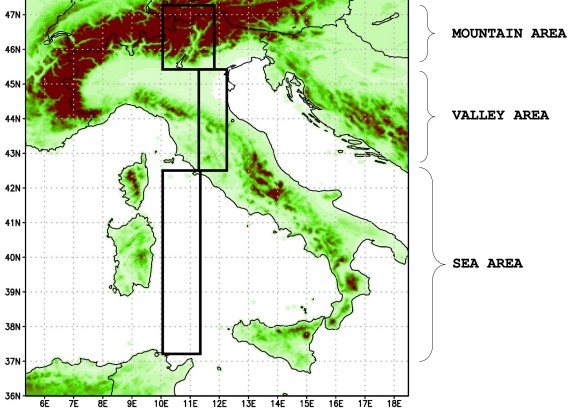

**Figure A3.** Selected areas (Sea, Valley, Mountain) used for the validation of the atmospheric fields of interest.

**Table A1.** RMSE and BIAS for the temperature at 2 m for the period from 18UTC of 24 September to 00UTC of 25 September 2020, and for the three selected areas (Sea, Valley, Mountain). The table shows the results of the comparison between the two ASSIM runs (Int: integrated PW, and Prof: WV vertical profile) and VR, and between CTRL and VR.

| | | \multicolumn{6}{}{Temperature 2 m} | | | | | |
|---|---|---|---|---|---|---|---|
| **Hour** | **AREA** | **Assim(Int)-VR** | | **Assim(Prof)-VR** | | **CTRL-VR** | |
| | | **RMSE** | **BIAS** | **RMSE** | **BIAS** | **RMSE** | **BIAS** |
| **18** | **Sea** | 0.4 | −0.4 | 0.4 | −0.4 | 0.4 | −0.4 |
| | **Valley** | 0.9 | −0.3 | 0.9 | −0.3 | 1.1 | −0.6 |
| | **Mountain** | 0.9 | −0.3 | 0.9 | −0.3 | 1.0 | −0.4 |
| **21** | **Sea** | 0.5 | −0.3 | 0.5 | −0.4 | 0.5 | −0.3 |
| | **Valley** | 1.0 | 0.1 | 1.3 | 0.4 | 0.8 | −0.2 |
| | **Mountain** | 0.9 | 0.0 | 1.1 | 0.3 | 0.9 | −0.1 |
| **00** | **Sea** | 0.5 | −0.4 | 0.5 | −0.4 | 0.5 | −0.4 |
| | **Valley** | 1.1 | −0.2 | 1.1 | 0.2 | 0.7 | 0.0 |
| | **Mountain** | 1.0 | −0.1 | 1.0 | 0.2 | 0.9 | −0.1 |

**Table A2.** As in Table A1, but for relative humidity (RH).

| | | \multicolumn{6}{}{Relative Humidity} | | | | | |
|---|---|---|---|---|---|---|---|
| **Hour** | **AREA** | **Assim(Int)-VR** | | **Assim(Prof)-VR** | | **CTRL-VR** | |
| | | **RMSE** | **BIAS** | **RMSE** | **BIAS** | **RMSE** | **BIAS** |
| **18** | **Sea** | 4.9 | 0.9 | 4.9 | 0.9 | 4.8 | 0.9 |
| | **Valley** | 7.0 | 1.4 | 7.0 | 1.4 | 7.6 | 1.3 |
| | **Mountain** | 7.1 | −0.9 | 7.1 | −0.9 | 7.3 | −0.3 |
| **21** | **Sea** | 6.5 | 0.1 | 6.2 | 0.0 | 6.1 | 0.7 |
| | **Valley** | 6.4 | −3.6 | 8.4 | −5.4 | 6.5 | −1.5 |
| | **Mountain** | 6.0 | −1.8 | 7.1 | −2.4 | 6.7 | −0.8 |
| **00** | **Sea** | 5.9 | 0.9 | 5.9 | 0.0 | 5.8 | 1.1 |
| | **Valley** | 10.1 | −6.6 | 7.2 | −5.7 | 4.5 | −3.1 |
| | **Mountain** | 7.6 | −2.4 | 7.3 | −2.4 | 6.0 | −1.1 |

**Table A3.** As in Table A1, but for accumulated precipitation, up to 3 and 6 h after assimilation (18UTC, 24 September 2020).

| | | \multicolumn{6}{}{Cumulated Precipitation} | | | | | |
|---|---|---|---|---|---|---|---|
| **Hour** | **AREA** | **Assim(Int)-VR** | | **Assim(Prof)-VR** | | **CTRL-VR** | |
| | | **RMSE** | **BIAS** | **RMSE** | **BIAS** | **RMSE** | **BIAS** |
| **18UTC +3 h** | **Sea** | 0.1 | 0.0 | 0.0 | 0.0 | 0.1 | 0.0 |
| | **Valley** | 5.9 | −1.1 | 4.5 | −1.8 | 4.4 | −1.2 |
| | **Mountain** | 10.4 | 0.8 | 10.1 | 0.3 | 10.8 | 0.5 |
| **18UTC +6 h** | **Sea** | 0.1 | 0.0 | 0.0 | 0.0 | 0.1 | 0.0 |
| | **Valley** | 5.5 | −0.5 | 3.8 | −1.0 | 4.1 | −0.7 |
| | **Mountain** | 7.8 | 0.7 | 7.0 | 0.1 | 9.0 | 0.7 |

The main elements that emerge from these quantitative comparisons can be summarized as follows:

1.  Overall, the assimilation after the first 3 h provides improved or in-line results in terms of BIAS and RMSE, primarily for the T2m variable for the three selected areas, and to a lesser extent for the RH variable, probably due to its high space-time variability.

2.  After 6 h from assimilation, however, the values returned line up or tend to be even worse than the CTRL ones for the three areas. This is probably due to the effects of assimilation beginning to dissolve quickly as time goes by, while the model tends to readjust itself according to its own dynamics and to the boundary conditions that prevail more often and faster in a relatively small simulation domain, as ours is.

3.  Regarding precipitation, the assimilation (in both cases) provides an improvement for the Mountain region, and slightly for the Sea in the first 3 h after assimilation. The Valley area is generally the most problematic, probably because it is less homogeneous, consisting of a mix of plains and Apennine reliefs. After 6 h, the benefits of assimilation still persist in terms of the lesser RMSE and BIAS, especially for the Mountain area. As shown in Figure A2, on average, over the mountainous areas, the values of NDSA measurements are higher than those of VR and CTRL, and this probably forces the model towards wetter atmospheric conditions and then towards greater probabilities of precipitation. Anyway, precipitation is a complex phenomenon that depends on several factors, such as temperature profiles, vertical instabilities and induced dynamics, advection, etc., so that deeper analyses are necessary to interpret such results.

Note that we have spoken about 'improvement' and 'worsening', but, in all cases, we are dealing with slight differences that are, however, coherent with the limited starting differences between VR and CTRL with respect to the errors in the assimilated data, and to the single-time single-transept assimilation procedure, that we have discussed above. In addition, it is worth stressing that the assimilation process in operational contexts is commonly achieved using different data types at different grid points of the simulation domain (from ground stations to radar and satellite data) in order to have a greater impact and induce a greater probability of forcing the forecast simulation towards a more realistic direction, avoiding worsening assimilation solutions. Here, instead, we are testing the impact of a single data type, and some worse results in the ASSIM run could probably be avoided just by assimilating some complementary observations, for instance, from heights lower than 2000 m. Concerning the wind field patterns, the variations (both in intensity and direction) between the VR and CTRL (bottom figures) and ASSIM of both the integrated PW values (central figures) and the WV vertical profiles (upper figures) were computed. These differences were calculated for the three selected areas and up to 6 h after assimilation. They are shown in wind-rose fashion, and, being differences, a perfect result should correspond to a single bar pointing towards north with all values equalling zero.

Figure A4 shows only the results for the Valley area for the case study of 24 September 2020, as it points out the greatest differences in terms of wind patterns, according to the complexity and heterogeneity of the studied area. The main elements emerging from these comparisons are summarized as follows:

1.  In general, the differences in intensity are comparable between the CTRL run and the two ASSIM ones in the first 3 h, then they slightly tend to worsen as time goes by, especially for the ASSIM with WV vertical profiles.

2.  Regarding the direction, the distribution of values in the ASSIM is quite in line with the CTRL, which already provides good results compared to VR in the first hours, with negligible improvement possibilities. A slight worsening occurs in terms of a broadening of the distribution of the direction as the hours pass since the assimilation.

As in the case of the previous meteorological fields, there is some dependence of the results on the characteristics of the selected area, and for the Sea and Mountain areas, the ASSIM results are very similar to those of the CTRL, already performing well with respect to VR, with a slight worsening as time passes.

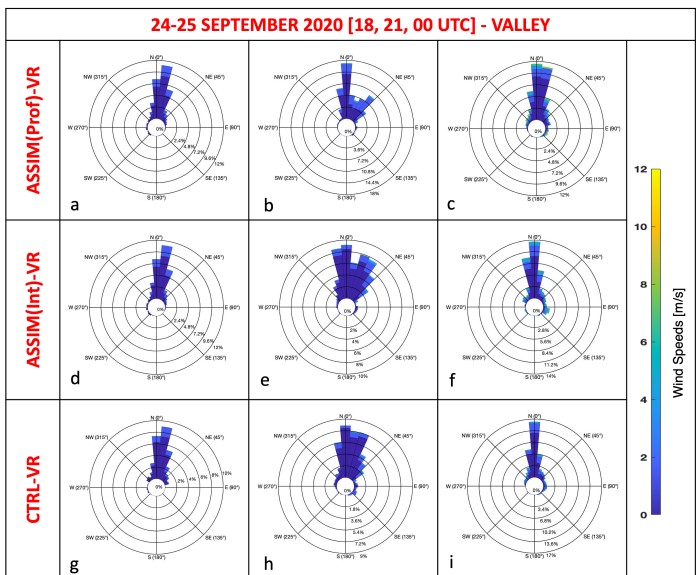

**Figure A4.** Differences in the wind field (direction and intensity) between the ASSIM runs (where Prof: WV vertical profile and Int: integrated PW) and VR and between CTRL and VR, for the case study of 24 September 2020 at 18UTC (subplots **a**,**d**,**g**), 21UTC (subplots **b**,**e**,**h**), and 00UTC of 25 September (subplots **c**,**f**,**i**), respectively. The results are for the Valley area.

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
