# Peer review of "Towards Space Deployment of the NDSA Concept for Tropospheric Water Vapour Measurements"

_atmosphere, doi:10.3390/atmos14030550_

Round 1

Reviewer 1 Report (Previous Reviewer 1)

The tropospheric water vapor measurements are very important to global climate and environmental studies. The authors presented detailed descriptions of the methods and the results. The messages obtained in the text are useful to the related communities. Overall, the article is well written. I suggest the authors double-check the contents and perform some minor changes, and the paper can be accepted for publication in the journal. 

Author Response

We have double-checked the contents of the manuscript and implemented minor changes to the text, according to the recommendation of Reviewer 1

Reviewer 2 Report (Previous Reviewer 2)

Dear Editor,

I am writing to recommend the acceptance of the revised manuscript, titled "Towards Space Deployment of the NDSA Concept For Tropospheric Water Vapour Measurements" for publication in Atmosphere.

As the reviewer for this paper, I have had the opportunity to review the initial submission and the revised version. The authors have made the necessary changes and improvements to the paper, addressing the comments and concerns that I raised in my review.

I believe that the revised paper is now ready for publication and makes a valuable contribution. The results of the research are novel and have the potential to impact future studies in this area.

I therefore recommend that the editor accept the revised paper for publication in Atmosphere. I am confident that it meets the high standards of the journal and would be a valuable addition to the publication.

Thank you for considering my recommendation.

Note, I prefer the second title, the authors leave the decision to the editor (see my first question).

“Towards Space Deployment of the Normalized Differential Spectral Attenuation Concept For Tropospheric Water Vapour Measurements”

Best regards

Author Response

We are grateful to the Revoewer 2 for the number of valuable comments and suggestions that substantially improved the quality and readibility of the manuscipt.

This manuscript is a resubmission of an earlier submission. The following is a list of the peer review reports and author responses from that submission.

Round 1

Reviewer 1 Report

Atmospheric water vapor, both the vertical distribution and column-integrated, is required in climate studies. In order to better determine the precipitation and radiation budget, we need to obtain high quality data of the atmospheric water vapor.

This article developed a novel measurement method. It can monitor the vertical distribution of the tropospheric water vapor. The method has now been demonstrated on theoretical basis, and is currently under development for space deployment.

In the manuscript, the authors provided all the details of the method, including theoretical considerations, physics and mathematical derivations, the calculation techniques and flowcharts. The messages and results are very useful to the relevant researches in the community. The paper was overall well written. I thus suggest acceptance of the paper for publication in the Atmosphere. The authors may perform some minor revisions.

Author Response

We performed some minor revisions, as recommended by Reviewer 1, in addition to major and minor changes explicitly suggested by Academic Editor and Reviewer 2.

Reviewer 2 Report

Review of the manuscript "TOWARDS SPACE DEPLOYMENT OF THE NDSA CONCEPT FOR TROPOSPHERIC WATER VAPOUR MEASUREMENTS" send to Remote Sensing Journal.

The authors demonstrate the normalized differential spectral attenuation technique that permits the estimation of the integrated water vapor along the radio link between a transmitter and a receiver carried by LEO satellites, focusing on the advancements resulting from the SATCROSS project. This paper deals with a delicate subject, the determination of water vapor in the troposphere (a complicated subject in itself), using recent LEO satellites. The problem is well explained, but the paper is too big. Therefore, I encourage authors to divide the subject into two papers. The first covers the technique and all the technical details that can be extended. The second addresses validation that can also be extended. In this way, the use of phrases such as " sake of synthesis" are avoided.

The paper is well written but very long. It has very long sentences and little punctuation; these points should be reviewed. I think the public of the Atmosphere Journal will not identify with this matter and advise before the Sensors Journal (MDPI). The assimilation section is confusing, and many results directly depend on it. For this reason, I advise major revision. The figures do not have publication quality; they must be revised, colors, text, organization, etc.

Title: avoid acronyms (like NDSA) in the title

L8: define SATCROSS

L13: improve the keywords; add "Normalized Differential Spectral Attenuation" for example

L16-10: add a reference

L21-22: format the references accordingly to the specifications of the journal, [1-4], [5-7]  

L24: radiosondes or probes

L33-34: add a reference to recent work about GNSS tomography "Miranda, P. M. A., & Mateus, P. (2021). A new unconstrained approach to GNSS atmospheric water vapor tomography. Geophysical Research Letters, 48, e2021GL094852. https://doi.org/10.1029/2021GL094852"

L38: reference [5] is not the best for this sentence. Add another one

L61: add a reference for each project ALMETLEO; ACTLIMB; ANISAP

section 2: board of two separate LEO satellites

L105: add a reference for ACE+ mission studies

L106: what do you mean by tropospheric turbulence

L174: move the reference [20] to the first occurrence of the project SWAMM in the main text

L176-187: did you do any signal analysis to conclude that the lousy signal part is due to multipath?

Figure 3: remove the dashed line box

Figure 4: what can cause the ripple that we can see for the horn antenna? add an identification letter to each subplot (a-e). You can also merge the first 2 plots and the second two with a second y-axis

Figure 5: do not have the quality to be published

L205: what is the mean rank of A

Equation 10: add a reference

L214: [25-26]

L215: [29-31]

Figure 7: what variable do we see in figure 7?

Figure 8: add a description for subplots (c) and (d)

L300: spin-up time used in the WRF model?

L303: The ecmwf forecast is known to be superior to the NCEP-GFS. The relatively small differences obtained were probably reduced to zero with the ecmwf forecast

Figure 11: add an identification letter to the subplots

L358-373: in fact, these conclusions depend on the assimilation approach, you probably used the CV3 covariance matrix error to perform the assimilation. This is not the correct way, you have to compute at least the CV5 matrix using one month of data and try again.

L383: You are getting wind differences because you use the CV3 matrix. Not to be used for regional domains.

L415: define CO-ROT

L521: to early to use the word “accuraty“ since this concept has not been demonstrated in this paper

L529: remove “developed”

L546: “greatest” -> “moderate”.

Rephare “The greatest impact  is evidently on the precipitation patterns and intensities, but also other meteorological variables such as temperature, 548 relative humidity and wind are affected.”. Since it depends on the assimilation approach.  

Table 1: Ora

Figure 14: settembre. Use another colormap and add a letter to identify each subplot

Figure 15, 16 and 17: merge into one figure, Figure 15a,b,c

Figures 17 and 18: check the y-label text

Author Response

Please, find the reply to Reviewer 2 in the attached file Reply_to_Reviewer 2.pdf
